# It Takes All of Us: How the Cystic Fibrosis Foundation Is Supporting States in Advancing Cystic Fibrosis Newborn Screening

**DOI:** 10.3390/ijns11020039

**Published:** 2025-05-20

**Authors:** Mary Dwight, Albert Faro

**Affiliations:** The Cystic Fibrosis Foundation, Bethesda, MD 20814, USA; afaro@cff.org

**Keywords:** cystic fibrosis, newborn screening, newborn, neonatal screening, CFTR, sequencing, immunoreactive trypsinogen, IRT, guideline, genetic testing

## Abstract

The publication of *Cystic Fibrosis Newborn Screening: A Systematic Review-Driven Consensus Guideline from the United States Cystic Fibrosis Foundation* (CFF) presents the challenge of implementation. CFF is prepared to partner with stakeholders to enhance newborn screening (NBS) practices. Through funding provided to the Center for Public Health Innovation (CPHI), the CFF has helped establish two genetic testing resource centers to help states implement CFTR sequencing within the NBS algorithm. CPHI, with CFF funding, is facilitating quality improvement collaboratives that unite CF clinicians and NBS staff nationwide to share best practices in laboratory methods, communication, and education. CFF continues to fund the Screening Improvement Program Award for Optimizing the Diagnosis of Infants and has developed a toolkit to help CF care teams collaborate with NBS programs on guideline implementation. Together, these initiatives aim to support states and CF providers in adapting their algorithms and processes. By identifying current best practices to improve timeliness, sensitivity, and equity in CF NBS, CFF seeks to promote better outcomes for all individuals with CF. Recognizing the competing demands on state public health departments, CFF is committed to partnering with stakeholders to ensure meaningful improvements in CF NBS.

We are pleased to see the publication of *Cystic Fibrosis Newborn Screening: A Systematic Review-Driven Consensus Guideline from the United States Cystic Fibrosis Foundation* [1]. The recommendations in this guideline describe the ideal state for cystic fibrosis (CF) newborn screening (NBS), and we recognize that implementation is a significant undertaking that will take time and resources. The CF Foundation is committed to supporting and partnering with CF clinicians and state NBS programs as they work to make changes to their algorithms and processes.

The widespread adoption of NBS for CF was a monumental step forward for our community and continues to be cause for celebration. Early diagnosis is essential for infants with CF, and through NBS, diagnosed infants can start treatment within the first month of life, leading to improved health outcomes. However, with the advances in screening methods for CF NBS, we see opportunities for continued improvement in timeliness and equity. This guideline provides a framework for states to optimize the NBS process and ensure that all infants have equal opportunity to be identified early in life.

For instance, implementation of next-generation sequencing (NGS) is resource-intensive, both in costs and in technical expertise. To assist with this, the CF Foundation is supporting two CF NBS Genetic Testing Resource Centers through a grant to the Center for Public Health Innovation (CPHI). These centers will be housed at the Wisconsin State Laboratory of Hygiene and the Wadsworth Center at the New York State Department of Health and will offer additional *CFTR* sequencing capacity for states that do not currently have the technology or expertise for NGS. These centers will also offer technical assistance for other states looking to perform their own *CFTR* sequencing. We hope these centers can alleviate some of the resource and knowledge challenges that surface as states consider adopting NGS to improve their screening sensitivity for CF.

We are also investing in partnerships between CF clinicians and state NBS programs, with the goal of helping states collaboratively map out potential improvements through data-driven decisions. We are doing this by supporting the CPHI quality improvement collaboratives that bring together CF clinicians and NBS staff from across the country to share best practices in laboratory methods, communication, and education. The CF Foundation also provides grant funding for quality improvement projects through the Screening Improvement Program Award for Optimizing the Diagnosis of Infants, which opened for another award cycle in May 2025. Finally, we have released a toolkit for CF clinicians with resources and tips for CF providers to work with their NBS programs to collaboratively assess their states’ performance and identify areas for improvement.

By identifying best practices to improve timeliness, sensitivity, and equity for all infants born in the U.S., we hope that implementation of this guideline will lead to improvements in health and well-being for people with CF and advance our mission to provide all people with the disease an opportunity to lead long, fulfilling lives. We recognize the many challenges state public health departments face in managing a broad range of programs beyond NBS. As the CF community continues to strive towards a better, more equitable landscape for NBS, we remain committed to supporting both CF care teams and NBS programs to help make this vision a reality.

## Data Availability

The original data presented in the commentary are openly available in the *International Journal of Neonatal Screening* at https://doi.org/10.3390/ijns11020024.

## References

[B1-IJNS-11-00039] McGarry M.E., Raraigh K.S., Farrell P., Shropshire F., Padding K., White C., Dorley M.C., Hicks S., Ren C.L., Tullis K. (2025). Cystic Fibrosis Newborn Screening: A Systematic Review-Driven Consensus Guideline from the United States Cystic Fibrosis Foundation. Int. J. Neonatal Screen..

