# Peer review of "It Takes All of Us: How the Cystic Fibrosis Foundation Is Supporting States in Advancing Cystic Fibrosis Newborn Screening"

_2409-515X, 2025, doi:10.3390/ijns11020039_

Round 1

Reviewer 1 Report

Comments and Suggestions for Authors

The authors emphasise the importance of this exercise (to develop clear standards for the implementation of bloodspot screening for CF), and the support of the CFF in the US to provide infrastructure to achieve these targets.

An issue that is not covered is how to ensure programmes comply.  A problem in achieving this remains the lack of consensus on rates of CRMS recognition and PPV targets (in particular).  Until the US has clear consensus on this metrics, it will be extremely challenging to implement change in the approach of screening labs.  There is a lot of wriggle room for the labs as it stands, to continue the status quo.

Author Response

Comment from Reviewer:

The authors emphasise the importance of this exercise (to develop clear standards for the implementation of bloodspot screening for CF), and the support of the CFF in the US to provide infrastructure to achieve these targets.

An issue that is not covered is how to ensure programmes comply.  A problem in achieving this remains the lack of consensus on rates of CRMS recognition and PPV targets (in particular).  Until the US has clear consensus on this metrics, it will be extremely challenging to implement change in the approach of screening labs.  There is a lot of wriggle room for the labs as it stands, to continue the status quo.

Response from authors: We appreciate this comment. The CF Foundation acknowledges that the guidelines it produces, including the CF Newborn Screening (NBS) guideline that is the focus of this commentary, are not mandates. It is up to state labs to decide whether to implement the published recommendations. We recognize that not all aspects of the guideline will be adopted immediately by every lab, but we hope over time that the evidence, the resources that the Foundation is funding (such as the Screening Improvement Program Award, NBS collaboratives, and the two genetic testing resource sites), in addition to innovation amongst the state labs will lead to improvements.

Reviewer 2 Report

Comments and Suggestions for Authors

Dear authors,

thank you for this commentary, which will surely help to implement the CF guidelines in the US.

However, I am sure there is world wide interest in the guidelines and also in the support that the CFF offers, for the US CF NBS.

Still it might also stimulate the improvement of CF NBS in other countries.

Author Response

Comment from reviewer: 

Dear authors,

thank you for this commentary, which will surely help to implement the CF guidelines in the US.

However, I am sure there is world wide interest in the guidelines and also in the support that the CFF offers, for the US CF NBS.

Still it might also stimulate the improvement of CF NBS in other countries.

Response from authors: Thank you.

Reviewer 3 Report

Comments and Suggestions for Authors

Nice summary of work of the foundation to support CF screening in the USA

Author Response

Comment from reviewer: Nice summary of work of the foundation to support CF screening in the USA

Response from authors: Thank you.